# *Salmonella* Infection in Poultry: A Review on the Pathogen and Control Strategies

**DOI:** 10.3390/microorganisms11112814

**Published:** 2023-11-20

**Authors:** Syamily Shaji, Ramesh K. Selvaraj, Revathi Shanmugasundaram

**Affiliations:** 1Department of Poultry Science, The University of Georgia, Athens, GA 30602, USA; syamily.shaji@uga.edu (S.S.); selvaraj@uga.edu (R.K.S.); 2Toxicology and Mycotoxin Research Unit, US National Poultry Research Center, Athens, GA 30605, USA

**Keywords:** *Salmonella*, vaccines, poultry

## Abstract

*Salmonella* is the leading cause of food-borne zoonotic disease worldwide. Non-typhoidal *Salmonella* serotypes are the primary etiological agents associated with salmonellosis in poultry. Contaminated poultry eggs and meat products are the major sources of human *Salmonella* infection. Horizontal and vertical transmission are the primary routes of infection in chickens. The principal virulence genes linked to *Salmonella* pathogenesis in poultry are located in *Salmonella* pathogenicity islands 1 and 2 (SPI-1 and SPI-2). Cell-mediated and humoral immune responses are involved in the defense against *Salmonella* invasion in poultry. Vaccination of chickens and supplementation of feed additives like prebiotics, probiotics, postbiotics, synbiotics, and bacteriophages are currently being used to mitigate the *Salmonella* load in poultry. Despite the existence of various control measures, there is still a need for a broad, safe, and well-defined strategy that can confer long-term protection from *Salmonella* in poultry flocks. This review examines the current knowledge on the etiology, transmission, cell wall structure, nomenclature, pathogenesis, immune response, and efficacy of preventative approaches to *Salmonella*.

## 1. Introduction

*Salmonella* is the leading cause of foodborne diseases worldwide that infects the gastrointestinal tract and causes diarrhea, nausea, and cramps in humans [1]. The Center for Disease Control and Prevention (CDC) estimates that approximately 1.35 million infections and 420 deaths are reported annually in the United States. *Salmonella enterica* ser. Enteritidis (*S.* Enteritidis), and *Salmonella enterica* ser. Typhimurium (*S.* Typhimurium), belonging to the non-typhoidal *Salmonella* group (NTS), is responsible for the majority of human salmonellosis. Globally, non-typhoidal *Salmonella* is responsible for approximately 93 million cases of gastroenteritis and 155,000 fatalities annually. The severity of human salmonellosis varies depending on factors such as the specific strain causing the infection, health conditions, and host age. It has been reported that the infective dose in a human infant is reported to be 100 bacterial cells, and even fewer cells are required to cause an infection in an immunocompromised individual [2,3,4].

Poultry serves as the main reservoir for various non-typhoidal *Salmonella* (NTS) serotypes among food-producing animals. Epidemiologically significant NTS serotypes include *S.* Typhimurium, *S.* Enteritidis, *S.* Heidelberg, and *S.* Newport. In North America and Europe, *S.* Enteritidis dominates the egg-borne transmission of infection to humans, whereas *S.* Typhimurium was the primary serovar associated with external egg contamination in Australia [5,6,7]. Between 1998 and 2008, poultry accounted for 17.9% of foodborne illnesses in the United States, with *Salmonella* ser. Enteritidis and Typhimurium are responsible for 17.4% and 34% of poultry-related foodborne illnesses, respectively [8]. In 2016, a national outbreak of multidrug-resistant *S.* Heidelberg linked to chicken products produced by a single poultry company was reported in California and Washington, leading to high hospitalization rates and indicating high virulence strains of *Salmonella* [9].

Society pays significant health costs and economic burdens caused by nontyphoidal *Salmonella* associated with chickens, estimated at 2.79 billion annually. This concern is growing as the global demand for ready-to-eat food products continues to rise [10]. Given the emergence of multi-drug-resistant bacteria and the associated public health crisis, alternatives to antibiotics are gaining more importance in the poultry sector, as antibiotic residues are known to contaminate consumed meat [11]. Several preharvest and postharvest intervention strategies have been developed to ensure the safety and hygiene of poultry products. Preharvest interventions include farm-level management, including the use of feed additives and biosecurity measures. Post-harvest methods include activities during carcass slaughter and meat processing, including the implementation of Hazard Analysis Critical Control Point (HACCP) plans [4].

Furthermore, with the FDA’s decision to abate the use of antibiotics in animal agriculture and consumer demand for antibiotic-free chicken, alternative control strategies that include vaccination programs are routinely followed by producers to combat pathogenic bacteria. Ensuring the sustainable production of poultry products is critical to meeting the future global demand for poultry products. However, annual *Salmonella* outbreaks can significantly impact production efficiency and food safety. To eradicate pathogens like *Salmonella*, which has multiple infectious serotypes, more powerful vaccines that confer cross-protection against multiple serotypes, including emerging serotypes, and induce long-term immunity are needed.

This review article focuses on the current understanding of *Salmonella* as a pathogen, its pathogenesis, and control strategies against *Salmonella* in poultry.

## 2. Etiology and Transmission

*Salmonella* spp. is a Gram-negative, oxidase-negative, non-spore-forming bacillus member of the Enterobacteriaceae family. *Salmonella* is a facultative anaerobic intracellular bacillus with a cell length between 2–5 µm. They are non-fastidious, motile, and have peritrichous flagella, except *Salmonella enterica* serovar Gallinarum and Pullorum. Most *Salmonella* serovars, except *S.* Typhimurium, are aerogenic. *Salmonella* is capable of producing hydrogen sulfide and converting nitrates to nitrites. Most *Salmonella* typically grows at temperatures ranging from 5–45 °C, with the ideal temperature being between 35–37 °C and at the optimum pH range of 6.5 and 7.5. Some strains can even grow at a pH as low as 3.7. *Salmonella* is sensitive to salt concentrations ranging from 0.5% to 5% and survives in environments with higher water activity (a_w_) ranging from 0.96 to 0.99. They can also survive in low-moisture food products for a long duration [12]. The severity of *Salmonella* infection varies according to many factors, including host age, host immunity, the presence of coinfections, environmental stress, managerial characteristics, and infective dose. Older birds, for instance, tend to be less susceptible to Salmonellosis even with concentrations of 10^6^ CFU/mL of *S.* Typhimurium [13].

*Salmonella* is distributed worldwide and is endemic to areas where animal husbandry is practiced. Serovars also vary in their distribution across the world, with ST and SE being prevalent everywhere. Some serovars are host-specific, like *Salmonella* ser. Abortusovis in sheep, *Salmonella* ser. Choleraesuis in pigs, and *Salmonella* ser. Dublin in cattle. Typhoidal *Salmonella serovars* like *S.* Typhi and *S.* Paratyphi are human pathogens transmitted via the fecal-oral route. In contrast, NTS is zoonotic and can infect a wide range of animal reservoirs, including birds, reptiles, dogs, cats, and rodents [14].

The primary reservoir for *Salmonella* in animals, particularly poultry, is the primary source of food-borne human salmonellosis. Transmission occurs mainly through the consumption of contaminated egg and meat products [12]. During the production cycle, poultry can become infected with *Salmonella* through various routes, including contact with carrier animals like rodents, cats, and insects. Contaminated poultry feed, litter, water, and aerosol transmission also contribute to the transmission of *Salmonella* [15].

*Salmonella* contamination of eggs can occur via two routes, namely horizontal and vertical, particularly by the serovar *S.* Enteritidis. In vertical or transovarial transmission, the infection occurs directly in the yolk, vitelline membrane, and albumen before the egg is laid. The infection originates in reproductive organs such as the ovary and oviduct with *S.* Enteritidis. As a result, bacteria enter the egg even before the eggshell is formed in the oviduct [16]. In horizontal or fecal-oral transmission, eggs become contaminated by eggshell penetration from the colonized gastrointestinal tract (GIT). Additionally, contaminated feces transfer the pathogen to eggs during or after oviposition. Feces serve as nutrient reservoirs for *Salmonella*’s growth, contaminating the environment and potentially infecting the rest of the flock in the same enclosure. The bacterial penetration of the egg is more rapid during the first few minutes post-oviposition, as some cuticles are immature and few pores are open [17]. Outer shell contamination by *Salmonella* is evident in eggs collected from contaminated nests and hatchery environments. Some studies reported that there is no direct relationship between eggshell thickness and *Salmonella* Typhimurium penetration, but eggs with high specific gravity shells tend to offer more resistance against *S.* Enteritidis penetration [16].

Insects can act as vectors for *Salmonella* on poultry farms. Cockroaches, for example, have the potential to introduce foodborne pathogens like *Salmonella* into poultry production facilities due to their ability to cross-contaminate and transmit the pathogen to uninfected individuals within their group. Studies have indicated that cockroaches infected with *S.* Typhimurium can transfer the bacteria to the surface of the table egg [18]. Flies captured in poultry establishments have been shown to harbor *Salmonella*. The poultry mite (*Dermanyssus gallinae*) has been implicated as a biological vector of *Salmonella* Enteritidis and has been reported to carry the bacteria within poultry premises. It is suggested that the primary source of infection could be oral ingestion of crushed contaminated mites by the chicks, as well as the mite’s blood meal [19]. *Alphitobius diaperinus*, also known as the litter beetle, has been found to transmit *Salmonella* to poultry in an experimental infection [20]. According to Bastiaan and Aize, rodents such as mice can act as carriers of *Salmonella* in layer flocks [21]. Feral mice present in poultry farms may serve as a rich source of multiple phenotypes and genotypes of *S.* Enteritidis [21]. In African countries, *S.* Kentucky and *S.* Enteritidis were the major serovars isolated from lizards and rodents inhabiting the poultry houses. It is hypothesized that the feces of lizards and rodents could contaminate the feed and litter, posing a biosecurity threat [22].

*Salmonella* can colonize the intestines of wild birds, turning them into asymptomatic reservoirs. Hughes et al. reported the isolation of different *Salmonella* serovars from wild birds, mainly passerine birds, in northern England [23]. *S.* Typhimurium strain DT160 caused significant mortality in wild birds and gastrointestinal illness in humans in New Zealand in 2000, indicating a zoonotic risk [24]. Wild birds like one buff-necked ibis, red-legged seriema, and eared dove captured near poultry facilities had *Salmonella* infection. Moreover, *S.* Typhimurium dominated the serotype isolated from wild pigeons [25]. It is hypothesized these birds play a vital role in the transmission of *Salmonella* serotypes to poultry and humans during migration, seasonal movements, and feeding [26]. Olga et al. found that 32.3% of the bacterial pathogens identified in the wild bird population in a national park in Ukraine tested positive for *S.* Enteritidis. These birds migrate to various parts of the world, contributing to the distribution of the pathogen to locations far from its source [27]. Furthermore, a shift in the poultry *Salmonella* serotypes linked with the spread of clones has been identified as a contributing factor to the increase in human cases of *S.* Infantis between 2011 and 2013 across Europe [28]. Likewise, *S.* Heidelberg, with its multiple clones in the US, is one of the top poultry and human serotypes associated with multistate outbreaks [9]. Figure 1 summarizes the overview of the transmission routes and vectors of *Salmonella* in poultry.

## 3. Cell Wall and Nomenclature

The cell wall of *Salmonella* consists of lipids, lipopolysaccharides (LPS), proteins, and lipoproteins [29]. The complex polysaccharide moiety with the Lipid A portion serves as an endotoxin and is responsible for bacterial virulence [29]. The LPS complex comprises three regions: (1) an outer O polysaccharide-specific side chain, (2) a middle core portion, and (3) an inner lipid A. The somatic O antigen is the central component of the endotoxin, composed of various monosaccharides and polysaccharides [29]. The Somatic O antigen is the polysaccharide portion on the bacterial surface that contains multiple [5,6,7,8] monosaccharides. The O-specific side chains determine the immunological specificities of the respective O-antigens [30]. Furthermore, all *Salmonella* O antigens share five common sugars: heptose, ketodeoxyoctonic acid, D-glucose, D-galactose, and D-glucosamine. It has been hypothesized that all-specific *Salmonella* polysaccharides have a common core composed of these sugars, with attached chains made of sugars unique to the specific serotype [31]. In summary, variations in the LPS structure exist in the O-polysaccharide chain, and these variations are responsible for the antigenic factors that confer serotype specificity.

Genotypic analysis indicated that *Salmonella* has about 60 somatic antigens using gene transfer [32]. Based on these antigens and specific antisera, *Salmonella* can be grouped into different serotypes. In brief, *Salmonella* has three cell surface antigens: an O antigen (cell-wall somatic), an H antigen (flagellar), and a K antigen (capsular). The flagellar antigen is a thermolabile protein, while the somatic O antigen is thermostable [4]. Some serovars of *Salmonella* also have a K antigen, which is a heat-sensitive carbohydrate. Among the *Enterobacteriaceae*, *Salmonella* is unique for having two distinct H antigens: phase 1 (specific) and phase 2 (non-specific) flagella antigens [31].

*Salmonella* was initially isolated in 1885 by Theobald Smith and Daniel Elmer Salmon from pigs infected with classical swine fever [33]. Currently, the nomenclature system recommended by the World Health Organization (WHO), namely supplement 2001 (no. 45) to the Kauffmann-White scheme, is followed for research on *Salmonella*. According to this terminology, the genus *Salmonella* includes two species, *Salmonella enterica* and *Salmonella bongori*, that belong to it based on the differences in 16-sRNA sequence analysis. Historically, the two species are further subdivided into six subspecies based on biochemical and phylogenetic properties and denoted by Roman numerals: *enterica* (I), *salamae* (II), *arizonae* (IIIa), *diarizonae* (IIIb), *houtenae* (IV), and *indica* (VI) [32,34,35,36]. Among the subspecies of *Salmonella*, *S*. *enterica* subsp. *enterica* is the one most commonly associated with human and animal infections [32].

However, the serovars within the subspecies are named according to the Kauffman and White classification system, which is based on the immunoreactivity of the major *Salmonella* cell surface antigens. Each *Salmonella* serovar possesses a unique antigenic combination, and a serovar (or serotype) name is assigned to each individual combination of the O:H1:H2 antigenic formula [37]. The Kauffman-White Scheme was first published in 1934 with 44 serovars. The latest one, released in 2007, has over 2500 serovars [38]. According to this classification system, the *Salmonella* serovars are formed by different combinations of 46 O antigens and 114 H antigens.

*S*. *enterica* serovars are the etiological agents of typhoidal and non-typhoidal *Salmonella* infection, with the former being host-restricted to humans and the latter zoonotic with a broad homeothermic host range [34]. *S. enterica* comprises over 2400 serotypes. Among the 100 serovars of epidemiological importance in the *S*. *enterica* species, *S.* Enteritidis and *S.* Typhimurium are the most prevalent serovars [39].

*Salmonella* serotypes are characterized and identified using various methods, including phage typing, PCR ribotyping, pulsed-field gel electrophoresis analysis, multi-locus DNA sequencing, and antimicrobial resistance patterns [40,41,42,43,44,45]. Recently, alternative molecular methods like the Check and Trace *Salmonella* assay (a microarray method using molecular markers) and whole-genome sequencing (WGS)-based serotype prediction tools have been used to characterize *Salmonella* isolates [46,47]. Compared to traditional serotyping, these methods are claimed to be easy, requiring only a short turn-around time. Traditional serotyping based on phenotypes can be challenging due to the complexity of typing sera, antigen preparation, and antisera required for testing [23,48].

## 4. Pathogenesis

*Salmonella* pathogenesis can be divided into several stages, including adhesion and invasion of gut epithelial cells, survival, multiplication within the host cells, and extraintestinal spread. *Salmonella*, being an enteric pathogen, reaches the intestine via oral ingestion (horizontal transmission) from contaminated environments, feed, and water. Even a very low infective dose of *Salmonella* Enteritidis, as low as 1–5 bacteria cells, can lead to infection in day-old chicks. The incubation period for *Salmonella* is usually 7 to 14 days [4]. The bacterium’s ability to withstand a pH of 3.7 in the stomach helps the bacteria pass through the acidic stomach environment [4].

Upon reaching the small intestine, *Salmonella* invades and adheres to the intestinal epithelial cells using fimbrial adhesins. *Salmonella*’s entry into the intestinal mucosa is facilitated mainly through M cells located over the Peyer’s patches (mucosal-associated lymphoid tissues). Other routes include internalization by dendritic cells and uptake by enterocytes mediated by effector proteins associated with virulence genes in SPI-1-TTSS [49,50].

M cells, specialized antigen sampling cells of the intestinal epithelium, play a critical role in the uptake and active transportation (transcytosis) of *Salmonella* to the underlying lymphoid follicles. This uptake of bacterial antigen by M-cells is vital in the development of mucosal and systemic immune responses, as *Salmonella* antigens are delivered to mononuclear phagocytes like dendritic cells (DCs) and macrophages [51].

Macrophages can internalize *Salmonella* but are unable to kill them as the bacteria can inhibit the fusion of phagosomes with secondary lysosomes, the mechanism used by macrophages to destroy intracellular pathogens. This enhances the intracellular survivability of the bacteria [52]. *Salmonella* proliferates inside the macrophages within a structure called the *Salmonella*-containing vacuole (SCV) and eventually translocates widely to the draining mesenteric lymph nodes, leading to bacteremia and invasion of systemic organs such as the liver, spleen, ovary, and gallbladder [52].

Both young and adult birds are infected by the invasive *S.* Enteritidis. Young birds tend to develop a systemic disease with high mortality rates, whereas adult birds can remain asymptomatic carriers post-colonization by the bacterial pathogen. The infection dose plays a critical role in developing clinical signs, with clinical salmonellosis being more likely to develop in young birds infected with high doses of *S.* Enteritidis [53].

Typhoid fever is caused by the bloodstream dissemination of *S*. *enterica* serotype Typhi [54]. Non-invasive serotypes of NTS are confined to the gastrointestinal tract, where they induce significant inflammation, including focal and diffuse mononuclear cell infiltration, necrosis of epithelial cells, edema, and eventually enterocolitis [55]. In brief, *Salmonella* has orchestrated mechanisms to co-evolve with their hosts by altering cellular processes that favor bacterial survival and intracellular proliferation.

*Salmonella* pathogenic factors are controlled by virulence genes and plasmids, which are located within *Salmonella* pathogenicity islands (SPIs). SPIs are a group of genes located in specific areas of the bacterial chromosomes that encompass multiple virulence factors, including invasins, adhesins, and toxins. *Salmonella* contains a total of twenty-three SPIs, with SPI-1 and SPI-2 being the two critical ones. Both of these SPIs encode a molecular apparatus called the type III secretion system (TTSS) or molecular syringe. This TTSS is responsible for injecting effector proteins produced by *Salmonella* into the host cell, thereby establishing the intracellular survival and propagation of *Salmonella* inside the host [55].

*Salmonella* has two types of type III secretion systems, T3SS-1 and T3SS-2, found in SPI-1 and SPI-2, respectively. Briefly, T3SS-1 is responsible for transferring effector proteins needed for bacterial invasion, SCV (*Salmonella* containing vacuoles) biogenesis, and inflammation. On the other hand, T3SS-2 facilitates the transport of effector proteins that favor SCV maturation, SIF (*Salmonella*-induced filaments) biogenesis, intracellular survival of the pathogen, and its movement inside *Salmonella*-containing vacuoles [56].

The effectors of SPI2-T3SS involved in SCV maturation and SIF biogenesis are SifA, SseJ, SopD2, PipB2, SseF, SseG, SpvB, and SteA. These effectors are essential for converting early SCV (endosomes) to late SCV, where bacterial replication occurs. *Salmonella*-containing vacuoles (SCVs) are specialized structures expressed by TTSS on the SPI-2 used by the bacteria to reside inside epithelial cells and macrophages to escape from killing by macrophages through the inhibition of lysosomal fusion [49].

*Salmonella*-induced filaments (SIFs) are specialized endosomal tubule projections that extend from the SCV. They are characterized by lysosomal glycoproteins and endocytic markers such as Rab4, Rab9, Rab11, and Rab5, among others. These SIFs inside the host cell cytoplasm form a complex replicative niche. The exact role of SIFs in *Salmonella* infection is still unknown [57]. It has been reported that the SifA-SKIP-Rab9 complex decreased M6PR (late endosomal/lysosomal markers) recruitment to the SCV membrane and reduced the movement of lysosomal enzymes to the SCV. This reduction in the movement of lysosome enzymes, expanding the SCV population, helps protect intracellular *Salmonella* from host defense mechanisms [58].

Further, studies have reported that *Salmonella* Typhimurium infection causes enterocytes to become M-cells, thereby exacerbating inflammation and the immune response associated with enhanced translocation of bacteria across the submucosa. This differentiation of the primed intestinal cells into M cells promotes *Salmonella* colonization and host invasion. The mechanism is controlled by the type III effector protein SopB via the Wnt/β-catenin signaling pathway [59].

A previous study reported by Yakhya et al. found that the deletion of SPI1 led to a decreased colonization of *Salmonella* Typhimurium in the cecum and spleen of chickens. They also reported that the deletion of SPI2-T3SS did not significantly affect the cecal colonization by *S.* Typhimurium. However, the SPI2 mutation reduced the ability of *S.* Typhimurium’s ability to invade the spleen in one-week-old chicks [60]. In addition, the Michael et al. study revealed that the disruption of *spaS*, an essential component of the SPI-1 T3SS, had little influence on cecal colonization in day-old chicks. In contrast, the mutant *spaS* strain was shown to reduce cecal colonization and decrease liver invasion during an experimental *S. Typhimurium* infection in one-week-old birds [61]. Moreover, the same study illustrates that the deletion of another gene, *ssaU*, that encodes major components of SPI-2 T3SS did not affect bacterial colonization of the intestine but exhibited a significant reduction in *S.* Typhimurium dissemination to the liver throughout the study [61]. A recent study with the *S.* Typhimurium challenge model showed that the deletion of SPI-1 and SPI-2 genes had a negative impact on the mutant strain’s ability to colonize and cause systemic lesions in the cecum and liver in one-day-old chickens [62].

*Salmonella* spp. has the ability to form biofilms at room temperature on surfaces in poultry environments and food processing plants. These biofilm cells are highly resistant to antimicrobials and contribute to the increased virulence of the bacteria, thereby establishing a chronic infection. The ability of *Salmonella* to survive in biofilm poses challenges for disinfection procedures in poultry environments [63]. Figure 2 summarizes *Salmonella* pathogenesis.

## 5. Immune Response to *Salmonella*

The interaction between humoral and cell-mediated immune systems plays a critical role in clearing *Salmonella* infection in poultry. The complex immune response against *Salmonella* depends on multiple factors, including host species, host age, gut microbiota composition, bacterial load, and *Salmonella* serotype associated with the infection [64,65].

### 5.1. Innate Immune System

The innate and adaptive immune components are the two main classifications of the avian immune system. The innate immune system serves as the first line of defense against pathogens, and it has several key components, including (1) chemical and physical barriers such as skin, mucosal epithelia, mucus, and antimicrobial molecules; (2) blood proteins such as complement, lectins, and agglutinins; (3) phagocytic effector cells (macrophages, neutrophils, monocytes), dendritic cells, heterophils, and natural killer cells; (4) cytokines; (5) cellular receptors (pathogen recognition receptors) such as Toll-like receptors, NOD-like receptors, and RIG-like receptors; (6) antimicrobial peptides such as defensins and cathelicidins [66].

The avian immune organs can be divided into bone marrow, bursa of Fabricius and thymus, and peripheral lymphoid organs such as the spleen and cecal tonsils [67,68]. MALT (mucosa-associated lymphoid tissue) comprises components of the immune system linked to the gastrointestinal and bronchial mucosa, providing the first line of protection against pathogens that enter the body through these mucosal surfaces. GALT, a major component of MALT, contains an isolated or group of lymphoid follicles disseminated in the lamina propria of the gastrointestinal tract, such as the bursa of Fabricius, cecal tonsils (CT), Meckel’s diverticulum, Peyer’s patches (PP), and intraepithelial lymphocytes. GALT lymphoid structures are the major secondary lymphoid organs since chickens lack other peripheral encapsulated lymph nodes [69,70].

Pathogen-associated molecular patterns (PAMPs) are microbial molecular structures that stimulate the innate and eventually adaptive immune systems. Some examples of PAMPs include nucleic acids such as single-stranded RNA (ssRNA) and double-stranded RNA (dsRNA); proteins such as flagellin; cell wall lipids such as lipopolysaccharide (LPS) and lipoteichoic acids; and carbohydrates like mannans found in fungi [66]. Cellular receptors of the innate immune system that recognize PAMPs in pathogens are known as pathogen recognition receptors (PRRs). PRRs are present in various immune cells, like phagocytes, dendritic cells, heterophils, and barrier epithelial cells. Toll-like receptors, NOD-like receptors, and RIG-like receptors are examples of PRRs. Studies have identified 10 Toll-like receptors in chickens [71]. The binding of PAMP to the specific TLRs results in the production of reactive nitrogen and oxygen intermediates, cytokines, and co-stimulatory molecules, initiating the adaptive immune response [72]. The outer membrane lipopolysaccharide (LPS) of Gram-negative bacteria is the ligand for TLR4, while TLR5 recognizes bacterial flagellin [66].

Okamura et al. reported that when birds were vaccinated with recombinant *Salmonella* flagellin (rFliC) and challenged with *Salmonella* Enteritidis, there was a significant reduction in bacterial load in the liver and cecum [73]. A transcriptome analysis study on the cecal mucosal immune system of layer chicks challenged with *Salmonella* Typhimurium demonstrated upregulation of TLR15, TLR1A, TLR2B, and TLR7 pathways on days 3, 5, and 7 post-infections with *S.* Typhimurium [74]. There was a measurable variation in TLR expression in multiple regions of the intestinal tract in 2-day-old broiler birds at 24 h post-challenge with *S.* Enteritidis. The infection upregulated the expression of TLR 1LA from the ceca to the ileum, TLR2A from the duodenum to the ceca, and TLR 15, which is unique to avians. However, there was a three-fold downregulation in the expression of TLR 5 compared to uninfected birds [75]. These findings emphasize the critical role of TLRs in the innate immune response of avians against *Salmonella*.

It has been reported that heterophils are integral cells of the avian innate immune system. They are granulocytic white blood cells that are equivalent to mammalian neutrophils and act as the first line of defense against pathogens in the avian immune system [76]. The functions of heterophils include phagocytosis of both opsonized and non-opsonized pathogens like *Salmonella*, degranulation of granular content, and the initiation of oxidative burst (ROS) [76]. Ferro et al. demonstrated upregulation of proinflammatory cytokines IL-6, IL-1, and IL-8 in heterophils of *Salmonella* Enteritidis-resistant one-day-old chicks compared to infected chicks [77]. Toll-like receptors, including TLR1, TLR 2, TLR 3, TLR 4, TLR 5, TLR 6, TLR 7, and TLR 10 mRNA, are associated with antipathogenic defense pathways in avian heterophils [78], and upregulation of TLR4, TLR15, and TLR 21 in heterophils following *Salmonella* Enteritidis stimulation [79]. A transcriptome analysis study has shown that the heterophil/ lymphocyte (H/L) ratio is a critical factor that determines the robustness of the immune response in birds [80]. A recent study reported that the number of goblet cells, IL-1, IL-8, and IFN-γ ileal expressions were all inversely linked with the H/L ratio, suggesting that birds with low H/L ratios had improved intestinal immunity and barrier functions [81]. In conclusion, these findings indicate the critical role of heterophils in defense against *Salmonella* and facilitate its clearance. However, further research is needed to determine whether the H/L ratio could serve as a biomarker to identify chickens resistant to *Salmonella*.

### 5.2. Adaptive Immunity

#### 5.2.1. CD4+ T-Cells, CD8+ T-Cells

Adaptive immunity plays a critical role in defense against intracellular pathogens like *Salmonella*. In brief, T cells are key immune cells in the cell-mediated immune response. Helper T-cells (CD4+ T-cells), cytotoxic/killer T-cells (CD8+ T-cells), and regulatory T-cells (Tregs) are functionally distinct populations of T lymphocytes [66]. CD4+ T cells orchestrate cytokine production and initiate the immune response by activating other immune cells. Their release of IL-2 and IFNγ by CD4 T cells stimulates NK cells, macrophages, and CD8-T cells and aids B cell differentiation into plasma cells. CD8+ T cells, also known as killer/cytotoxic T cells, cause the lysis or apoptotic cell death of intracellular pathogens [66].

Withanage et al. reported that a Th1-mediated decrease in the bacterial load was correlated with an upregulation in the number of CD4+ and CD8+ T cells in the spleen and liver of one-week-old Rhode Island Red chickens infected with *S.* Typhimurium [82]. Infection with *Salmonella* Enteritidis in laying hens has been shown to increase the numbers of both CD4+ and CD8+ T cells along with macrophages in the ovaries and oviduct around 7–14 days post-inoculation, which was followed by peak upregulation in B cells. The reduction in the bacterial recovery rate from the ovary and oviduct around the peak lymphocyte period indicated the involvement of cell-mediated immunity [83].

Moreover, Saenz et al. recently reported that an increased CD4+/CD8+ ratio in vaccinated birds challenged with three different *Salmonella* serovars was linked to a reduction in cecal bacterial load [84]. At one week post-*Salmonella* Enteritidis inoculation, CD4+ and CD8+ T cells significantly increased in the intestines of vaccinated birds compared to unvaccinated and non-challenged birds [85]. In addition, another study reported an increase in intraepithelial cytotoxic CD8+ T cells in SE-infected birds during multiple time points during the study. However, the number of splenic cytotoxic CD8+ T-cells and splenic helper CD4+ T-cells was comparable between the infected and uninfected birds across the course of infection. Thus, prevention of systemic clearance of pathogens from the spleen alone may be insufficient [86]. Research has shown that at seven days post-infection with SE, the CD11+ MRC1LB+ macrophage level increased in the spleen, where the bacterial load was the highest [86]. The results are supported by a transgenic chicken study that showed increased MHC-II expression levels in MRC1LB+ macrophages. This altogether suggests the importance of increased levels of macrophages and CD4+ and CD8+ immune cells in clearing *Salmonella* infection in chickens [87].

#### 5.2.2. Regulatory T-Cells

Regulatory T cells, or T-regs, are a subset of Th cells. Chicken T-regs (CD4+ CD25+) develop in the thymus and serve to maintain tolerance and suppress the immune response by producing the anti-inflammatory cytokine IL-10 [66]. IL-10 functions by inhibiting the production of IL-2 by effector T cells, which in turn suppresses the development of an IFNγ-driven proinflammatory response against pathogens. A study with *Salmonella*-challenged broiler chickens reported a significant increase in the number of T-regs and IL-10 mRNA expression throughout the study [88]. They also found that the T-regs from infected birds suppressed T cell proliferation on days 7 and 14 post-inoculation. Moreover, Tregs from uninfected control birds did not suppress T cell proliferation [88]. In conclusion, an increase in the Treg percentage was associated with persistent *S.* Enteritidis infection in the cecal tonsils of broiler birds [88]. Persistent *S.* Enteritidis and *S.* Heidelberg infections were reported in chickens by the induction of CD4+CD25+ cells and by the variation in IL-10 mRNA transcription, resulting in an asymptomatic carrier state in birds 18 days after infection [89]. In vaccinated birds post-challenge with *S.* Enteritidis, there was a decrease in the cellular immune response, explained by the significantly higher levels of IL-10 at the same time point [90]. Overall, these findings indicate that Tregs have an impact on *Salmonella* infection in chickens.

#### 5.2.3. Cytokines

The cells can be classified into Th1 and Th2 cells based on the types of cytokines they produce. Th1 cells promote the proinflammatory cell-mediated immune response against intracellular pathogens like *Salmonella* by producing interferon-gamma (IFNγ), IL-12, and tumor necrosis factor (TNF). Th2 cytokines, mainly IL-10, are anti-inflammatory, suppressing the Th1 response, and are associated with immune resistance to *Salmonella* and *Salmonella* carrier states in chickens [91].

In brief, exposure of ligands to PRRs on antigen-presenting effector cells like macrophages, dendritic cells, and B-cells causes the activation of APC, which in turn activates the naïve T-cells. The activated T-cells differentiate into various subsets of T-helper cells [66]. In addition, the production of IL-12 by activated macrophages, known as classical activation, induces the production of IFNγ by T cells. The production of IFNγ further induces the differentiation of T cells into Th1-type cells, marking the initiation of a cell-mediated immune response [66]. Earlier studies have shown that splenic IFNγ mRNA expression was higher in day-old chicks exposed to *Salmonella* [92]. The clearance of *Salmonella enterica* serovar Typhimurium from the spleen is associated with an increase in T-cell proliferation and IFNγ expression [93]. Another pro-inflammatory cytokine, IL-1β, was related to an acute immune response against *Salmonella* in young chicks [94]. A study demonstrated that IL-1β mRNA expression was increased in the ilea and cecal tonsils of *S.* Typhimurium-inoculated birds [92]. Furthermore, a transcriptome analysis study on the cecal mucosal immune system of layer chicks challenged with *S.* Typhimurium found a steady upregulation in the differentially expressed genes where IL-6 played a key role [74]. A study on the effects of killed *S.* Enteritidis vaccination in chickens showed increased IFNγ and IL-2 production by spleenocytes on antigen recall responses post-vaccination [95]. In birds immunized with a killed chitosan nanoparticle vaccine, the mRNA expression of TNF-α in cecal tonsils increased 0.5-fold when birds were inoculated with *Salmonella.* They also reported an increase in the expression of IL-17 in the cecal tonsils of immunized-challenged birds [96]. However, to date, no studies have reported the specific function of Th-17 cells in the defense against *Salmonella* in poultry.

Chausse et al. reported that genes linked to IFN alpha/beta were inhibited in birds resistant to *Salmonella*. In addition, they found that both carrier-state-resistant and susceptible birds had decreased IFNγ gene expression, suggesting suppression of the Th1 response [91]. *Salmonella* has developed a “unique survival strategy” for long-term colonization of the chicken intestine, resulting in persistent *Salmonella* infection and the development of a cecal carrier state. It was suggested that carrier status could be associated with a Th2 response, as indicated by the upregulation of cytokines like IL-4, IL-5, and IL-13 mRNA expression [91].

#### 5.2.4. B-Cells and Immunoglobulins

B cells play a crucial role in the humoral immune response by producing antibodies. These B cells differentiate into antibody-secreting plasma cells, which produce various Ig isotypes depending on the cytokine present. For instance, IgA serves as the first line of defense against mucosal infections in MALT and is the most prevalent antibody on mucosal surfaces [66]. The exact role of antibodies in controlling *Salmonella* remains unclear [97]. However, recent studies have identified that humoral immunity is also important in that the enteric pathogen decreases the production of serum IgG by B cells as a mechanism to escape from the humoral immune response. Manne et al. demonstrated that *Salmonella* uses the protein SiiE to specifically decrease the number of IgG-producing plasma cells in bone marrow, which in turn reduces serum IgG titer [98]. In addition, they also observed that the SiiE-depleted strain of *Salmonella* had increased serum anti-*Salmonella* IgG [98]. There have been reports of a steady increase in the antigen-specific serum IgG and IgM antibodies in birds inoculated with *Salmonella* Typhimurium [99]. In a study on chickens looking at the impact of chemically induced B cell depletion on immunological response, it was found that the birds with B cell deficiency had higher rates of intestinal *Salmonella* Enteritidis shedding [100]. Other studies claim that the clearance of *Salmonella enterica* serovar Typhimurium infection in birds is independent of the humoral immune response. They showed that B-cell-depleted birds had the same response as control birds to clearing the *Salmonella* infection [101]. Withanage et al. reported that intravenous inoculation of SE in layer birds led to peak production of IgG, IgM, and IgA antibodies around 14 days post-inoculation, correlating with the decrease in *Salmonella* recovery rates during the same period [94]. However, further research on the role of the humoral immune response in the chicken immune system is needed to get a clear understanding of the efficiency of B-cell and Ig repertoires in controlling *Salmonella* infection.

## 6. *Salmonella* Control Strategies in Poultry

### 6.1. Biosecurity

Implementing good biosecurity measures plays a key role in combating the transmission of *Salmonella* and improving food safety [102]. Biosecurity controls include entry-level and site cleanliness, immunization, boot dips, and the hand hygiene of employees. In addition, better rodent and fly control, red mite control, and disinfection in between flocks are recommended to reduce the incidence of Salmonella and to halt the disease cycle in farms [103]. To reduce the prevalence of D. gallinae, a biological vector of Salmonella, appropriate biosecurity measures should be used [104].

Proper litter management has been associated with a decreased risk of Salmonella detection in poultry houses. Moreover, higher Salmonella contamination was linked to the use of fresh wood shavings than older litter [105]. Hence, proper recycling of litter using methods like composting is crucial in reducing the Salmonella count in poultry litter [106]. Additionally, the use of proper disinfectants is vital in limiting the introduction and dissemination of disease in birds [107]. Furthermore, the seroprevalence of Salmonella increased during the summer compared to the winter season [108]. Hence, strict biosecurity measures are required to combat the seasonal prevalence of Salmonella among poultry flocks.

### 6.2. Antibiotics

Antibiotics have been used in poultry feed since the 1940s, mainly due to their benefits on birds’ feed efficiency, enhanced growth performance, and inhibition of enteric pathogens [109]. The list of antibiotics used as feed additives to combat enteric pathogens includes small quantities of penicillin, tetracycline, and chloramphenicol [110]. However, the subtherapeutic use of antibiotics in poultry feed is being reconsidered because of the growing concern about antibiotic resistance in the human food chain [111]. Resistant Salmonella serotypes have been reported against antibiotics such as quinolones, chloramphenicol, and cephalosporins worldwide [112]. Moreover, control strategies like probiotics are given importance as a viable substitute for antibiotics in light of the European Union’s (EU) ban on their usage and the US’s restricted use of them in the production of chickens. Additionally, the use of antibiotics is associated with the destruction of beneficial gut bacteria that help fight enteric pathogens [113]. Hence, alternatives to antibiotics like probiotics, prebiotics, synbiotics, postbiotics, etc., are given more importance in the post-antibiotic era.

### 6.3. Prebiotics

Prebiotics and probiotics play important roles in promoting gut health and supporting the balance of beneficial intestinal flora in poultry. According to the International Scientific Association for Probiotics and Prebiotics, prebiotics can be defined as “a substrate that is selectively utilized by host microorganisms conferring a health benefit” [114]. The Food and Agricultural Organization (FAO) of the United Nations (UN) defined probiotics as a “non-viable food component that confers a health benefit on the host associated with modulation of the microbiota” [115].

The prerequisite for a potential prebiotic includes the ability to withstand hydrolysis by gastric acids and enzymes and resist absorption in the upper gastrointestinal tract [116]. An ideal prebiotic should also be metabolizable by the gut microbiota, be a selective compound that promotes the growth of beneficial intestinal flora, and have the capacity to regulate the immune response in favor of the host while suppressing pathogens, thereby improving the host’s health and performance [117].

Prebiotics like non-digestible oligosaccharides and polysaccharides have been shown to inhibit the survival and colonization of pathogens like *Salmonella* by producing SCFA like butyrate and acetate in the ceca, which helps lower the gut pH [118]. Yeast cell wall-derived mannan oligosaccharides (MOS) [119], fructo-oligosaccharides (FOS), inulin [120], and xylo-oligosaccharides [121] are some examples of prebiotics used in poultry production systems to control various pathogens, including *Salmonella*.

The composition of the yeast cell wall includes MOS (40%), β-glucan (60%), and chitin (2%) [122]. Studies have demonstrated the benefits of yeast cell product supplementation in poultry diets. Shanmugasundaram et al. reported that yeast cell wall product supplementation in layer birds reduced fecal and intestinal oocyst count, up-regulated anti-inflammatory cytokine IL-10 production, and increased proliferation of beneficial bacteria like LAB and their by-products of fermentation in the cecal tonsils during post-coccidial challenge [123]. Another study found that whole yeast cell prebiotic supplementation in the diet of broiler birds modulates the immune response by increasing the percentage of Tregs, improving IL-10 (anti-inflammatory) mRNA expression, and reducing the pro-inflammatory cytokine (IL-1) mRNA expression in the cecal tonsils of broiler birds [124]. Supplementing broilers with MOS (0.05%) and FOS (0.25%) significantly improved overall body weight gains [125]. Dietary FOS supplementation at 1% revealed a reduction in the cecal load of *Salmonella* Enteritidis, an upregulated ileal IgA cell titer, and increased expression of TLR-4 mRNA in layer birds challenged with SE [126]. Broiler birds supplemented with 5% trehalose and inoculated with *Salmonella* Typhimurium improved feed conversion ratio, favored the growth of lactobacilli in jejunum and duodenum, decreased the cecal load of ST, and decreased inflammation in GIT [127]. These results highlight the potential benefits of supplementing poultry diets with prebiotics as an alternative to antibiotic growth promoters for controlling harmful bacteria like *Salmonella*, ultimately improving production performance and gut health.

### 6.4. Probiotics

Probiotics, also known as direct-fed microbial (DFM), are defined by FAO as “live microorganisms, when administered in adequate amounts, confer a health benefit on the host” [128]. Lilly and Stillwell first used the term and defined probiotics as “growth-promoting factors produced by microorganisms” [129]. In this context, probiotics confer their beneficial effects on the host through competitive exclusion, improving barrier health and function, immunomodulation, and digestion and absorption, thereby promoting growth and performance. The qualities of potential probiotic candidates are (1) host origin; (2) non-pathogenic and beneficial to the host by adhering to the gut mucosa (biofilm formation); (3) tolerate of gastric acid and bile salts; (4) antimicrobial properties against pathogens; and (5) survive post-processing and storage stress [130,131]. Probiotic microorganisms used for poultry supplementation include spore-forming *Bacillus* spp. *Saccharomyces* yeast, *Enterococcus* spp. [132], *Streptococcus* spp. *Lactobacillus* spp. and *Bifidobacterium* spp. [133]. The available probiotics on the market include either single-species or multispecies preparations, with the latter preferred due to its ability to act on multiple sites to bring out an overall synergistic effect [134].

The administration of probiotics to layers of birds improved egg production, egg weight, and egg quality [135]. Colonization by the *Bacillus subtilis* CSL2 probiotic strain normalized the level of fecal microbiota and increased *Lactobacillus* in *Salmonella*-challenged Hy-line Brown laying hens [136]. Supplementation of probiotics containing *Lactobacillus fermentum* and *Saccharomyces cerevisiae* in broilers improved feed efficiency and the percentage of intestinal T-lymphocytes (CD4+ and CD8+) [137]. Continuous supplementation of *Bacillus*-based probiotics restored the gut microbiota, decreased *Salmonella* load in the internal organs, and increased the level of butyrate in free-laying birds challenged with *S.* Typhimurium infection [138]. Probiotic supplementation has also been associated with increased anti-*Salmonella* bile IgA in birds challenged with *Salmonella*, indicating improved humoral immunity [139]. It has been shown that feed supplementation with *Lactobacillus* spp. and *Bifidobacterium* spp. has been shown to increase the production of proinflammatory cytokines IFN-γ and TNF-α that favor the clearance of *Salmonella* from the gut [140]. Moreover, studies have shown that probiotic bacteria play a role in maintaining a balance between pro-inflammatory cytokines and anti-inflammatory cytokines [141]. Recent studies have reported that the combination of probiotics (*Lactobacillus* spp., *Pediococcus* spp., *Saccharomyces* spp., and *Bacillus* spp.) with the live SE vaccine enhanced the growth performance, decreased mortality rate, and reduced fecal shedding of bacteria in *Salmonella*-challenged broilers, thereby limiting the bacterial colonization of birds [142]. These findings suggest that probiotics could be a potential alternative to antibiotics in poultry infected with *Salmonella* since they positively modulate the gut microbial population and enhance the immune response against the pathogen.

However, more research needs to be performed to determine the appropriate storage and packaging conditions for probiotics and to investigate the development of antimicrobial resistance in the gut microbial community. Some studies suggest that probiotics may deteriorate when exposed to temperatures higher than room temperature [143].

### 6.5. Synbiotics

Synbiotics refers to the synergistic combination of prebiotics and prebiotics, a concept first used by Gibson and Roberfroid in 1995 [144]. In synbiotics, prebiotics are used to support and sustain the probiotic microorganisms by modifying the gut microflora, ultimately enhancing the ability of probiotics to survive and inhibiting colonization of the gut epithelium by pathogens. The supplementation of synbiotics has shown significant benefits to host animals compared to using prebiotics and probiotics separately [145]. The Food and Agriculture Organization (FAO) recommends using the term synbiotics only if the combined health effect is synergistic [115]. Examples of synbiotics include combinations like fructo-oligosaccharides with bifidobacteria and lactitol with lactobacilli [146].

Synbiotic supplementation has been documented to improve the production performance and alleviate the heat stress of broiler breeders. Birds fed synbiotics showed less heat stress behaviors than birds fed a regular diet [147]. Supplementation of synbiotic preparations containing *Lactobacillus* spp., *Saccharomyces cerevisiae* yeast, and inulin has shown a positive effect on the proliferation of beneficial intestinal bacteria like *Bifidobacterium* spp. and *Lactobacillus* spp. in broiler birds [148]. Shanmugasundaram et al. investigated the effect of synbiotic supplementation on the production performance, immune parameters, and cecal *Salmonella* load of *Salmonella*-challenged layer birds. They reported improved body weight gain, higher hen-day egg production both with and without a *Salmonella* challenge, decreased cecal *Salmonella* colonization, and increased bile anti-*Salmonella* IgA [149]. In addition, synbiotics may help to modulate lymphoid organs (bursa, spleen), increase the size of bursal follicles, and stimulate the immunoglobulins, thereby improving immunocompetence against *Salmonella* Typhimurium infection in broilers [150]. These findings suggest that synbiotics have promising effects and could serve as potential growth promoters in poultry production.

### 6.6. Postbiotics

Postbiotics are non-viable bacterial products or metabolic byproducts, either secreted by live bacteria or derived after cell lysis from probiotic microorganisms, that confer beneficial functions on the host. In general, postbiotics range from SCFA to enzymes, organic acids (propionic and 3-phenylacetic acid), peptides, plasmalogens, vitamins, teichoic acids, and muropeptides. Postbiotics mimic probiotics in their mode of action, except that they are not alive [151]. The soluble factors are obtained from probiotic microorganisms in the cell walls and cytoplasm, such as *Lactobacillus* spp., *Bifidobacterium* spp., *Bacillus* spp., and *Saccharomyces cerevisiae*. Postbiotics benefit the host by producing immunomodulatory effects, decreasing gut pH, inhibiting pathogenic bacteria in the gut (pathogen antagonism), enhancing antioxidant properties, enhancing gut health, protecting intestinal barrier integrity, and improving production performance [152]. The most common postbiotics used are metabolites and their combinations produced by strains of *Lactobacillus plantarum* [153].

In the context of poultry, research has shown positive effects of postbiotics on various aspects of health and performance. In broiler birds exposed to heat stress, the postbiotic supplementation obtained from *L. plantarum* RI11 at 0.6% (*v*/*w*) showed positive effects on genes related to gut barrier health and reduced the expression of heat shock protein 70 genes as well as the acute phase protein genes, indicating the antioxidant function of postbiotics [154]. These birds also exhibited increased growth performance, intestinal histomorphology, a higher number of cecal microflora, and a negative effect on the cecal *Salmonella* population following the addition of postbiotics produced by *L. plantarum* in broiler birds maintained under heat stress conditions [155]. Dietary inclusion of postbiotics RG14 (0.15% and 0.45%) along with prebiotics (inulin) at a concentration of 1% exhibited improved body weight gain, a higher proportion of *Bifidobacterium* in the cecum accompanied by decreased mRNA expression of interleukins LITAF (Lipopolysaccharide-induced tumor necrosis factor-alpha factor) and IFN and reduced pathogenic bacteria like *E. coli* in broilers. Nevertheless, the diet did not affect the *Salmonella* population [156]. Choe et al. reported that supplementing layer birds with soluble metabolite combinations of *L. plantarum* strains improved egg production performance, fecal population of LAB, and intestinal villi height and crypt depth [157]. Additionally, postbiotics derived from *Saccharomyces cerevisiae* fermentation reduced the cecal colonization of Salmonella in broilers and layer pullets, making them a potential preharvest intervention to enhance food safety and improve production performance [158,159].

As antibiotic resistance becomes a growing concern, postbiotics are a potential alternative for combating enteric pathogens like *Salmonella.* They can play a valuable role in preharvest intervention to enhance food safety practices and improve production performance in poultry. However, further research is needed to better understand the mechanism of action and safety of postbiotics in the poultry sector.

### 6.7. Phytobiotics

Phytobiotics, also known as phytogenics or phytochemicals, are biologically active compounds obtained from plants used in animal production as feed additives because they offer health benefits and promote growth in animal production, including poultry. The bioactive substances derived from plants include saponins, flavonoids, terpenoids, and alkaloids [160]. These compounds are reported to have antioxidant, antiviral, antimicrobial, anticoccidial, anti-parasitic, immunomodulatory, anti-inflammatory, and endocrine stimulatory activities [161,162]. Dietary supplementation of poultry feed with garlic powder [163], clove and cinnamon [164], peppermint powder [165], and ginger [166] has been shown to improve overall production performance, feed conversion ratio, and body weight gain. Other examples of plants used as phytobiotics include black cumin, turmeric [167], calendula, oregano, green tea, and fennel, among others [168].

Supplementing a phytobiotic named Intebio in the diet of growing birds challenged with *S.* Enteritidis was reported to decrease the earlier inflammatory response via the downregulation of IL6, IL8L2, CASP6, and IRF7 at day 23, thereby limiting the colonization of the pathogen [169]. Ziheng et al. found that oregano essential oil (OEO) supplementation in drinking water could inhibit and treat infection by *S.* Pullorum and *S.* Gallinarum in commercial yellow-chicken breeders. Additionally, they reported that OEO was more effective in preventing infection than the treatment [170]. A diet containing 40 mgmL^−1^ to 80 mgmL^−1^ of garlic extract revealed antimicrobial properties in broiler chicks challenged with *S.* Typhimurium by reducing mortality and improving body weight [171]. Phytogenic compounds like *trans*-cinnamaldehyde and eugenol have decreased *S.* Enteritidis growth and cecal colonization in challenged broiler birds after ten days of infection [172]. Moreover, natural capsaicin derived from chili pepper has been reported to control the internal organ (liver, spleen) invasion by *S.* Enteritidis in challenged layer birds [173]. All these studies indicate the great potential of phytobiotics as an antimicrobial substitute against *Salmonella* in poultry and its application in commercial farms.

### 6.8. Bacteriophages

Bacteriophages are viruses that infect bacteria and use the host machinery to proliferate inside the host cell. These phages penetrate their DNA into the host cell (lysogenic) and undergo multiplication, followed by the release of a large number of new bacteriophages, eventually leading to the lysis of the bacterium to release the progeny bacteriophages. In the lysogenic cycle, phage DNA integrates into the bacterial chromosome and can remain dormant for some time without causing cell lysis [174]. Bacteriophages are used as alternatives to antibiotics due to their promising target specificity, less allergic side effects, and harmlessness to the host’s normal flora [175]. The treatment of *S.* Enteritidis-contaminated poultry carcass with a higher number of bacteriophages was able to reduce the percentage of recoverable bacteria by 93% [176].

The application of phage cocktail (F1055S, F12013S) as an aerosol spray on fertile eggs challenged with *S.* Enteritidis during their transfer from incubators to hatchers reduced *Salmonella*’s horizontal transfer [177]. The inoculation bacteriophage cocktail made from phages isolated from chickens (UAB_Phi20, UAB_Phi87) and pigs (UAB_Phi78) decreased the *Salmonella* load in the cecum and mortality rate of white leghorn birds challenged with *S.* Typhimurium [178]. The administration of CTCBIO phage significantly reduced the *S.* Enteritidis load in the cloacal swab, liver, and spleen in broilers that are challenged with *S.* Enteritidis [179]. Broad-host-range phage (STP4-a) is beneficial over specific phages since they can inhibit multiple serovars of *Salmonella* with their polyvalent adsorption sites [180]. Additionally, oral inoculation of *S.* Enteritidis and *S.* Typhimurium phages reduced depression, loss of appetite, and diarrhea in a *Salmonella* challenge model. There is a significant decrease in the cecal colonization of *Salmonella* 7–15 days post-administration of phages in infected chicks [181]. In pandrug-resistant *S.* Typhimurium infected chicks, a phage combination (virulent and non-productive) enhanced the survival rate of chicks by 100%. It decreased the bacterial load in internal organs but did not improve body weight gain, alleviate splenomegaly, or re-establish the intestinal microbiota [182]. These data suggest that bacteriophage treatment could improve the survival rate of *Salmonella*-infected chickens and reduce the bacterial colonization of internal organs. The poor efficacy of phage to withstand the acidic gastric pH of the birds on oral delivery is overcome by the encapsulation technique [183]. However, the main disadvantage of phage therapy is the emergence of phage resistance [184].

### 6.9. Vaccination against Salmonella in Poultry

#### 6.9.1. Live-Attenuated Vaccine

Live attenuated vaccines, as the term suggests, are vaccines with living bacterial pathogens that have been rendered inactive or avirulent using attenuation methods such as chemical and genetically engineered mutagenesis [185,186]. These vaccines mimic natural infection as they adhere to the intestinal mucosa when administered orally, eliciting potent humoral and cell-mediated immune responses [185]. In newly hatched chicks with immature immune systems, *Salmonella* live vaccines promote resistance to infection, inhibiting gut colonization or competitive exclusion [187].

In a study by Lin et al., a live attenuated bivalently lyophilized vaccine containing a final concentration of 6 × 10^8^ CFU and 1 × 10^8^ CFU ST and SE strains, respectively, was inoculated into commercial layers at day 5, week 8, and week 18 of age [188]. The birds in different treatment groups were separately challenged with SE and ST at 25 weeks of age, and the birds were humanely euthanized on day 14 post-challenge. Both vaccines successfully eliminated cloacal *Salmonella* shedding, as observed from cloacal swabs collected on days 7 and 14 post-challenge. Triple vaccination with the ST vaccine significantly reduced the invasion of *Salmonella* into the internal organs like the liver, spleen, and cecum in 90% of vaccinated birds [188].

Groves et al. showed that oral administration of a live-attenuated *aro*A deletion mutant *S.* Typhimurium vaccine (Vaxsafe ST) at a concentration of 10^8^ CFU/250 µL decreased *Salmonella* load only for a temporary duration over a period of 17–25 weeks. However, the dual administration of the same vaccine through the parenteral (subcutaneous) route enhanced the cecal clearance of the bacteria [189]. In a study performed on the aroA mutant *Salmonella* Typhimurium live vaccine, there were no reductions in the fecal shedding of ST [190].

Many live attenuated vaccines are produced by mutating genes involved in crucial virus survival and metabolism [191]. A previous study explored the potential of an O-antigen-deficient live attenuated *Salmonella* Typhimurium vaccine, created by the deletion of the *rfaL*, *lon*, and *cpxR* genes, on the level of protection from systemic colonization and immune response. The vaccine induced significant ST-specific IgY and rapid clearance of mutant bacterial strains from the spleen and liver in about seven days [191].

Venessa et al. immunized day-old laying–type chicks with a live *Salmonella* Enteritidis/Typhimurium bivalent oral vaccine, followed by two booster immunizations at six weeks and 16 weeks of age. The birds immunized with the bivalent vaccine (*Salmonella* Duo) decreased the spleen colonization of a heterologous strain of bacteria when the birds were challenged with 4.6 × 10^8^ CFU of *Salmonella* Infantis (a heterologous strain). Hence, this study has been shown to confer cross-protection against *S.* Infantis (serotype C strains) [192]. However, despite the ability of live-attenuated vaccines to stimulate both cell-mediated and mucosal immune responses, the biosecurity risk of virulence reversal of the live vaccine strain is the major drawback of live-attenuated vaccines [193].

#### 6.9.2. Killed or Inactivated Vaccine

Killed bacterin vaccines made from inactivated whole-cell preparations of bacteria have been extensively used to control poultry *Salmonella* infection [194]. Commonly used inactivation agents for the preparation of killed vaccines include heat (60 °C for 1 h), formaldehyde, acetone, ethylene oxide, beta-propiolactone, and radiation (ultraviolet or gamma) [195]. The available *Salmonella*-killed vaccines are serovar-specific. Studies have shown that killed vaccines confer humoral immunity mostly and do not elicit strong cell-mediated immune responses. Therefore, booster vaccination is required for long-term protection [196,197]. Despite the absence of robust cell-mediated immunity, killed vaccines are preferred over live vaccines due to biosecurity and safety reasons [198]. Additionally, multivalent inactivated vaccines are required to contain the spread of a wide range of *Salmonella* serovars present in poultry [199,200].

An inactivated aluminum hydroxide-gel adjuvanted trivalent *Salmonella enterica* vaccine was shown to reduce the load of *Salmonella* Enteritidis in challenged birds by four log CFU/g, as well as demonstrate a complete reduction in bacterial dissemination to the liver when inoculated with *Salmonella* Infantis in vaccinated birds [194]. It has been shown that greater levels of IgA and IgY antibodies were produced in the bile and serum following the simultaneous use of live attenuated and killed *Salmonella* vaccines in broilers [201]. The combined use of both live and killed vaccination programs successfully reduced *S.* Typhimurium and *S.* Infantis colonization in young layers, providing broad protection [199].

#### 6.9.3. Subunit Vaccine

Subunit vaccines, made of defined antigens, are used in poultry and are claimed to be safer than live-attenuated or inactivated vaccines [193]. Vaccines derived from the outer membrane proteins (OMPs) and flagella proteins (FliC protein) of *Salmonella enterica* serovar Enteritidis with adjuvants have been used to decrease bacterial shedding in poultry and to induce a significant antigen-specific immune response against *Salmonella* [202]. Additionally, a trivalent subunit cochleate system-based vaccine has been evaluated against three serovars of *Salmonella* in layers. This vaccine increased serum IgY and improved production performance [84].

Desin et al. showed the efficacy of a subunit vaccine based on the Type 3 secretion system encoded on a type 1 pathogenicity island, which showed an increased titer of IgG antibodies in birds. Bacterial colonization in the liver was reduced but not in the spleen or cecum [203]. Renu et al. experimentally designed an oral mucosal adhesive biodegradable chitosan *Salmonella* subunit nanoparticle (CNP) vaccine for broilers, which successfully elicited mucosal IgA production, lymphocyte recall assays, serum IgY antibody levels, and conferred cross-protection against other *Salmonella enterica* serovars. Further, there was a significant reduction in *Salmonella* Enteritidis load in the ceca and spleen of CNP-vaccinated birds at 21 days post-inoculation [204].

In an experimental vaccine challenge study performed to study the protective effect of the vaccine on *Salmonella* serotypes, it was shown that a direct correlation exists between the increase in serum antibodies and a decrease in bacterial loads in the intestines of vaccinated birds [84]. The vaccine in the study did not affect the production performance of the vaccinated birds. Nevertheless, the duration of the protective response and the ability of these vaccines to confer cross-protective immunity are still unclear. However, not many vaccine studies on layer performance post-vaccination are available.

#### 6.9.4. Ghost Vaccine

In the past, *Salmonella* ghost vaccines have been extensively tested in rat models [205]. Bacterial ghosts are dead bacterial cell structures retaining surface antigenic LPS components without cytoplasmic contents made from Gram-negative bacteria. The lysis of Gram-negative bacteria mediated by the phage protein E produces bacterial ghosts [206]. A *S.* Enteritidis ghosts (SEGs) vaccine made by chemically mediated lysis was tested in rats challenged with virulent *S.* Enteritidis to assess its efficacy and capability to confer immune protection. They reported a significant increase in serum IgG antibody levels in SEGs-vaccinated rats [205].

Moreover, further studies have studied the safety of the ghost vaccines, as the E-gene-mediated lysis process, driven by the osmotic pressure-led bacterial cell emptying, is reportedly inefficient [207]. Chetan et al. studied the efficacy of the *S.* Enteritidis (SE) ghost vaccine, SE-LTB ghost, and a commercial vaccine. They found rapid clearance of the bacterial load in all groups post-SE challenge and an absence of local tissue reactions at the injection site. They also observed an increased level of IgG titer in SE-LTB-immunized birds [208].

More recently, *S.* Enteritidis ghost vaccines have been produced and adjuvanted with ST flagellin antigen (ST FliC). Birds immunized with the ghost vaccine demonstrated significant clearance of the SE wild-type challenge strain from the spleen and liver, increased serum IgY antibody levels, and an improved cell-mediated immune response [209]. Together, the above data indicate the paramount importance of further research into ghost vaccines as potential *Salmonella* vaccine candidates. However, it is important to note that since they are non-living cells, they cannot penetrate the gut epithelium and trigger a potent mucosal immune response, which could be a potential challenge when using the ghost vaccine to control *Salmonella* in chickens.

*Salmonella*, an intra-phagosomal enteropathogenic bacterium, has been used as a carrier for poultry DNA vaccines. DNA vaccines are considered safer than live vaccines because they are based on genes of antigenic proteins ligated to a plasmid, which is then transformed into attenuated *S.* Typhimurium and administered to birds [210]. The capacity of attenuated *S.* Typhimurium to elicit strong mucosal and systemic immune responses within the host is a beneficial feature for using it as a live carrier to combat a range of poultry viral infections [211,212]. Some advantages of DNA vaccines include no risk of infection and minimal interference with passive maternal antibodies [213]. DNA vaccines can be administered by intramuscular, oral, and in ovo routes in poultry [214]. Despite all its benefits, concerns about integrating DNA into the host genome and the development of anti-DNA antibodies are the central claims raised against DNA vaccines [213]. Figure 3 summarizes the control strategies against *Salmonella* in poultry.

## 7. Conclusions

With the increasing global demand for poultry meat and egg products, ensuring safe and hygienic poultry management is critical. A multi-intervention strategy that reduces the *Salmonella* bacterial load in birds and ultimately prevents carcass contamination in processing plants should be given more importance in the current era of multi-drug-resistant *Salmonella*. At present, there are a plethora of options claiming to reduce *Salmonella* in poultry rearing, but it is critical to exercise caution before implementing any of those approaches on a large scale. Additionally, continuous research should be performed to ensure the safety of all available novel control strategies. Further efforts should be made to study alternative vaccination strategies that can promote long-lasting immunity and understand the various virulence mechanisms of the zoonotic bacteria that can impair the effectiveness of the immune response induced in the birds.

## Figures and Tables

**Figure 1 microorganisms-11-02814-f001:**
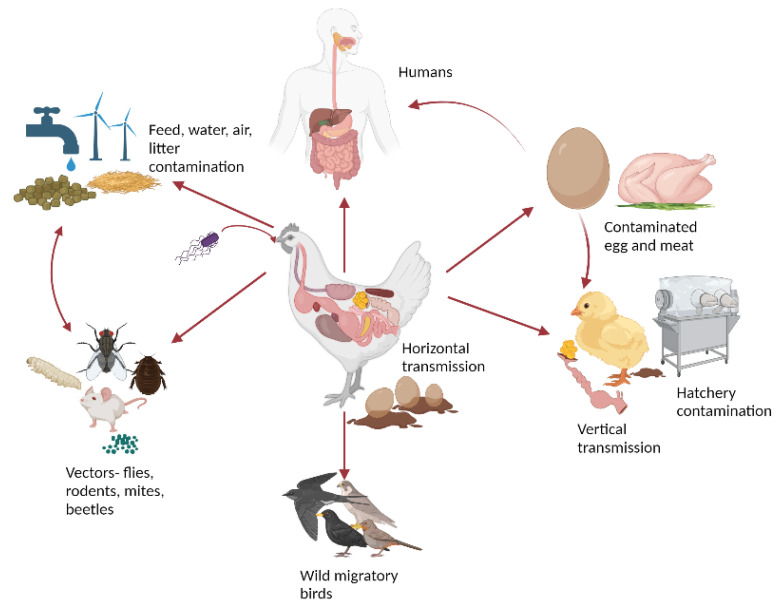
Overview of the various transmission routes of *Salmonella*. Created with Biorender.com (accessed on 26 July 2023).

**Figure 2 microorganisms-11-02814-f002:**
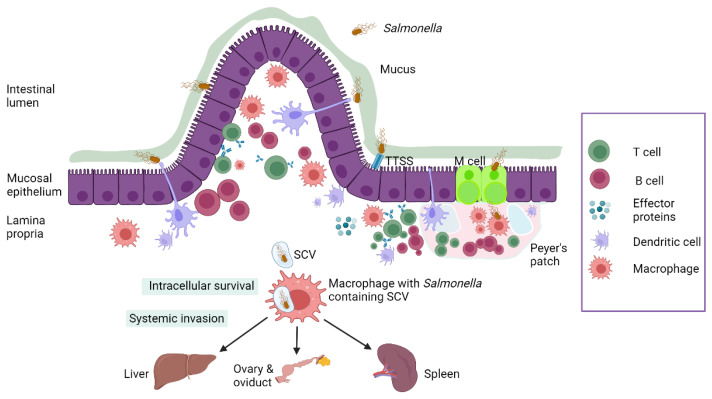
Schematic representation of *Salmonella* pathogenesis in poultry. Created with Biorender.com (accessed on 26 July 2023).

**Figure 3 microorganisms-11-02814-f003:**
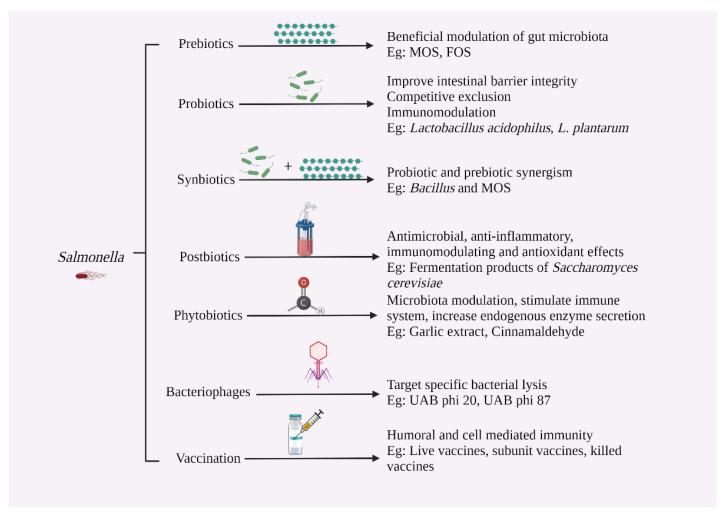
*Salmonella* control strategies in poultry. Created with Biorender.com (accessed on 26 July 2023).

## Data Availability

No new data were created or analyzed in this study. Data sharing is not applicable to this article.

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
