# Peer review of "Salmonella Infection in Poultry: A Review on the Pathogen and Control Strategies"

_microorganisms, 2023, doi:10.3390/microorganisms11112814_

Round 1

Reviewer 1 Report

Comments and Suggestions for Authors

As it is a review, the text is very long and detailed, providing chronology and basic definitions of occurrence and pathology. However, topics 5 and 6 are well updated, enriching the review.

Some parts of the text are all in italics, such as on pages 3 and 4.

The image in Figure 2 and 3 needs to be relocated to have good visibility on the page.

The conclusion can be improved, as there are punctuation problems (excessive use of semicolons in the text, as well as the absence of a full stop at the end). The final paragraph could emphasize the most recent and relevant aspects, already presented throughout the review, but not present in this topic.

Reviewer 2 Report

Comments and Suggestions for Authors

1. The presentation of relevant data is supported by literature or reports. For example,’

Center for Disease Control and Prevention (CDC) estimates that approximately 1.35 million infections and 420 deaths are reported annually in the United States.’, ‘Globally, non-typhoidal Salmonella is responsible for approximately 93 million cases of gastroenteritis and 155,000 fatalities annually. T’

2. Section 6.8 should be included in 6.7.

3. Salmonella control strategies should also include other modalities such as disinfection of farm premises, antibiotic application, etc.

Reviewer 3 Report

Comments and Suggestions for Authors

The authors have submitted a review article that outlines current knowledge and research gaps regarding Salmonella Infection in poultry. The review of currently available scientific literature about Salmonella Infection in poultry is of great importance from the standpoint of public, scientists and poultry production.

The title, abstract and keywords accurately reflect the content of the manuscript. The authors of this manuscript gave us a clear introduction based on the currently available scientific literature in this filed. Other sections in the manuscript are clearly defined and well discussed. In the Conclusion section authors summarised the main points regarding Salmonella Infection in poultry. References consist of appropriate and relevant papers.

According to my opinion, the manuscript should be accepted for publication in Microorganisms Journal as it stands.
